# How to Improve the Market Penetration of New Energy Vehicles in China: An Agent-Based Model with a Three-Level Variables Structure

**Mo Chen** [1,*] **, Rudy X. J. Liu** [1,2] **and Chaochao Liu** [1,2]

1    School of Management, Tianjin University of Commerce, Tianjin 300134, China; jliu@tjcu.edu.cn (R.X.J.L.); liuchaotjcu@tjcu.edu.cn (C.L.)
2    Research Center for Management Innovation and Evaluation, Tianjin University of Commerce, Tianjin 300134, China
*    Correspondence: chenmo@tjcu.edu.cn; Tel.: +86-13752705715

**Abstract:** This paper develops an agent-based model with linking variables (ABML) to investigate the influencing factors for the new energy vehicles (NEVs) market in China. The ABML is a framework with three-level variables including micro, linking, and macro variables, which can reduce the complexity of the simulation. The emergence from bottom to top occurs between linking and macro variables, while the best–worst scaling describes the mapping between micro and linking variables. In the case study, Rookie, Veteran, and New Generation consumers are assumed as the three types of consumers in China's market. A specification of the three types of variables is presented, where the value of linking variables obeys uniform distribution. By introducing the population density and the interaction frequency, the number of agents is determined with an experiment. All parameters in the model are estimated by calibrating the realistic vehicle sales. We compare different scenarios and obtain some management insights for improving the market penetration of NEVs in China.

**Keywords:** new energy vehicles; agent-based model; linking variables; China; market penetration

## 1. Introduction

New energy vehicles (NEVs), including battery electric vehicles (BEVs) and plug-in hybrid electric vehicles (PHEVs), have broad development prospects in China, but their current market share is relatively low. By the end of 2020, the number of traditional internal combustion engines (ICEs) is over 270 million, still dominating the market. The usage of NEVs is one form of sustainable consumption [1]. The concerns over climate change and limited energy have spurred policymakers to consider various solutions to the complex and intertwined environmental, energy, and transportation problems facing China. In response to the rapid development, local firms and joint ventures both begin to plan their NEV layouts. The car-sharing business is a typical case.

Although the State Council has issued the NEV product planning since 2012, the NEV ownership is about 4.9 million by 2020, with a low-level market share of 1.75%. For mitigating purchase anxiety, the Chinese government not only exempts vehicle purchase tax, vehicle, and vessel tax but also strengthens traffic and purchase restrictions of ICEs. These policies have contributed to the annual sales of more than 1 million from 2018 to 2020. The growth rate of NEV sales is up to 30% in 2020, and the sales are primarily from those restriction purchase cities, such as Beijing, Shanghai, and Shenzhen. In contrast, ICE sales are still dominant in many no-restriction purchase cities.

The prospect of the NEV market is ambiguous and involves many thorny problems. First, the percentages of different class NEV sales fluctuate every year. The percentage of A-class PHEVs is down to 32% in 2020, compared with 49% in 2019. The percentage of C-class PHEVs is up to 36% in 2020, compared with 15% in 2019. The percentage of A-class BEV is down to 34% in 2020, compared with 57% in 2019. The percentage of A00-class

BEV is up to 32%, compared with 23% in 2019. The market fluctuation is caused by some competitive NEVs, such as the Model 3 from TESLA and the MINIEV from SGMW. Second, the purchase intention of NEVs may be exaggerated. It is reported in some consultations that the public is very interested in purchasing NEVs, but the sales indicate that many potential buyers are hesitant to purchase NEVs [2]. The solution to this problem relies on a nexus of multiple influences: electric range, the availability of charging stations, and infrastructure environment, etc. [3]. Third, NEV strategies are diversified among different automobile enterprises. Some famous independent brands, such as BYD and BJEV, focus on the BEV market. Some luxury brands, such as Benz, BMW, and Audi, focus on the PHEV market, while some famous automobile enterprises, such as Toyota and Honda, focus on the Hybrid Electric Vehicles (HEVs) market. HEVs have achieved a good balance between low fuel consumption and nice power, but it is not on the NEVs list in China.

Developing NEVs needs enormous investments, including building infrastructures, reconfiguring resources, and formulating related policies. Thus, it is an important issue to analyze the influence factors of market penetration for decreasing the investment risk. China has kept the high growth of vehicle sales for over 10 years and NEVs will play an increasingly important role in the future. Some researchers generally establish models with many influence factors, which inherently limits the validity of sensitivity analysis. This paper innovatively proposes a general research framework for solving the dilemma between the number of variables and the emergence of an agent-based model (ABM). The ABM, with a three-level variable structure, which includes knowledge (micro variables), association (linking variables), and conception (macro variables), will help us simulate the purchase process of NEVs in China, explore the interaction among three types of consumers, and analyze the effects of different policies.

The paper is organized as follows: Section 2 presents the literature review in the scope of the study and the research areas of ABM. Section 3 gives a specification of model description. Section 4 presents a detailed case study in the number of agents, calibration, verification, results, and discussions. Section 5 presents the managerial and theoretical implications. Section 6 presents conclusions and future research directions.

## 2. Literature Review

### 2.1. Scope of the Study

Lim [4] provides a blueprint for future sustainability marketing studies, including economic, environmental, social, ethical, and technological dimensions. The NEV market analysis is more correlated to the economic dimension, and its research methods mainly include statistical analysis and model simulation. The former prefers statistical methods and choice models based on real data or investigation results. Melton et al. [5] summarize the PHEV-supportive policies in 10 Canadian provinces and provide an accessible framework to assess the effectiveness of both demand-focused and supply-focused policies, such as financial incentives, carbon pricing, emission standards, etc. Under the analysis, the most effective policies are Zero-Emission Vehicle mandates, strong and long-duration financial incentives, and strong taxation on gasoline. The application condition relies on adequate consumer data and a mature NEV market, and thus the framework is not suitable for China's market. Capturing common policy incentives, Langbroek et al. [6] set some attributes to cover different types: range, price after subsidies, public charging, parking benefits, and use of the bus lane. Following different hypotheses, they compare five models with mixed logit models and consumer survey data. However, the large number of product portfolios, generated by orthogonal fractional factorial design, will reduce the validity of the utility maximization framework. Wang et al. [7] use data mining combined with deep-learning technologies to investigate a large number of purchasing reasons and use correlation analysis to examine the relationship between the sales and the factors.

The methods of model simulation have three types: system dynamics models, diffusion, and time series models and agent-based models [8]. The system dynamics model must predetermine an overall causal loop diagram with passing validity tests and show

good conformity in the real system [9]. It has the advantage in better nexus stability for using differential equations. Its disadvantage is in describing complex interactions, and thus the predetermined loop diagram is not suitable for the immature NEV market in China.

Time series models have the advantage in implementing and fitting historical trends and are not valid in simulating the diffusion of a product where there exists a competitor [10]. The ultimate market potential for each vehicle must be estimated outside of the model. Of the erratic fluctuation of the NEV market, the application of time series models has some drawbacks in China's NEV market.

ABMs have the advantage in simulating agents' interactions with individual characteristics, needs and preferences, and have the disadvantages in verifying the high complexity and uncertain sensitivities [11]. The philosophy of ABM is a bottom-up approach exploring the connection between the micro-level behavior of individuals and the macro-level patterns. The more the micro variables are, the more precise the model is. Thus, how to deal with too many variables is the key to using ABM.

## 2.2. Research Areas of ABM

This study on NEVs using ABM is closely related to four research areas: model variable, agent types, flow charts, and mixed models.

### 2.2.1. Model Variable

Model variables include demand-side, supply-side, and policy-side variables [12]. Demand side contains consumer survey data, empirically based preferences, consumer awareness, and heterogeneous preferences. Supply-side variables contains electric range, charging infrastructure, fuel price, and so on. Policy-side variables have two types: demand-focused variables and supply-focused variables. The former attracts consumers by purchase subsidies, rollout of charging infrastructure, and free parking. The latter uses Zero-Emissions Vehicles and research subsidies to stimulate vehicle manufacturers and fuel suppliers.

### 2.2.2. Agent Types

According to the requirements of the model, there are many options for the agent types. Sullivan et al. [13] propose an agent model with four types of agents: consumers, governments, fuel producers, and vehicle producers. They can choose from 12 models of vehicles produced by 3 original equipment makers and decide whether or not to buy a car in each cycle. Cui et al. [14] integrate a consumer choice model into the virtual automobile marketplace and model spatial distribution of PHEV ownership at the local residential level with a case study on Knox County, Tennessee. Spangher et al. [15] develop an agent-based stochastic simulation with multiple vehicle types and apply the output to quantify the effect on the greenhouse gas emissions. Kangur et al. [16] explore how policies interact with consumer behavior over such a long time period. Through comparing the frequency of different agents in various references, the consumers are the preferred agent types.

### 2.2.3. Flow Chart

If–then rules are the core of the flow chart, which describes different agents' behaviors. Eppstein et al. [17] construct a nine-step flowchart to simulate annual vehicle updates, and introduce an agent attribute G to weigh the perceived benefits related to saving gasoline. Weighing the vehicles by geographic neighborhood and social network, the interactions use the agents' threshold T as the media. Shafiei et al. [18] develop a vehicle choice algorithm to make a purchase decision by multiple iterations in Iceland, which has rich domestic renewable energy. By generating a random variable with a uniform distribution, the interactions are reflected by comparing the social group's likelihood of purchasing a vehicle. They predict the market share of ICEs and EVs during the period 2012–2030 and show that decreasing EV prices and sufficient recharging infrastructures are

the key factors. Wolinetz et al. [19] develop a Respondent-based Preference and Constraint model to forecast the PHEV market share, in which the consumer's discrete choice is the core theory. The cycle among three sub-models forms the interactions, which includes constraints model, choice model, and vehicle model. Silvia et al. [20] give a specification of the agents' attributes and use eight questions to establish the decision rules depicted by a complicated flowchart. Randomly assigned environmental attitude score and innovation score describe the results of the interactions among agents in four distinct geographic sections.

### 2.2.4. Mixed Models

Some relevant theories can be introduced to expand the application scenario of ABM. McCoy et al. [21] compare two different social networks including small world and preferential attachment and simulate the adoption of EVs among Irish households. They find that mild peer effects could result in large clusters of adopters forming in certain areas, and it is a good way to put pressure on electricity distribution networks. Noori et al. [22] establish the EVReMP model, namely, a combination of some different methodologies. They test the word-of-mouth effect and predict that the electric vehicle market share will up to 30% in 2030 on average. Hesselink et al. [23] present an agent-based model on the adoption of energy efficiency by households, including technologies model, policies simulation, decision theories, and empirical data. The mixed ABM becomes a new trend.

In summary, current research focus on how to build the flowchart of ABM. In addition to the consumer choice model, the social network theory and the comprehensive model are also enriching the research in this field. Too many variables and complicated interactions will decrease the stability of emergence, but little research pays attention to the tradeoff between granularity and accuracy. Although decomposition is a common method for a large-scale ABM depending on a specific background, the relationship among these sub-models is rarely discussed. Simplifying interaction rules is another solution for sacrificing the accuracies of the model, but the larger granularity of the model is unmatched with China's NEVs market. Considering the lack of a systematic way to solve the dilemma between the micro level and macro level, this paper will introduce linking variables as a bridge to establish a general framework of ABM.

## 3. Model Description

### 3.1. Agent Choice

Potential consumers are chosen as agents in this research, and they are categorized into three types: Rookie, Veteran, and New Generation. The categorization is newly developed and suitable for China's market. Since 2009, the vehicle sales of China are the top in the global market, while the rapid development of vehicles is uneven in some regions with different economic levels, policies, and sales. These conditions form different cognition levels of consumers in vehicles, and the characters of consumers are summarized [24].

Rookie, denoted as NV1, refers to consumers who have not purchased vehicles, and lack use experiences, vehicle knowledge, and personal preference. Veteran, denoted as NV2, includes consumers with clear personal preferences who have owned a vehicle and have more purchase experiences and vehicle knowledge. New generation, denoted as NV3, refers to consumers with open minds who have not purchased a vehicle and have a few use experiences and vehicle knowledge.

### 3.2. Agent Variables

The variables can be categorized into three types: micro variables, linking variables, and macro variables. The variable structure has better stability, as shown in Table 1.

**Table 1.** Agent variables.

| Variable Type | Variables | | | | |
|---|---|---|---|---|---|
| Macro variables | Market penetration, NEV purchase index (NEVP) | | | | |
| Linking variables | Traffic restriction index (TR) | Purchase restriction index (PR) | NEV Product competitiveness index (PC) | Government subsidy index (GS) | Infrastructure index (INF) |
| Micro variables | Traffic restriction time of ICE | The lottery probability for license plate of ICE | Battery warranty policy | Exempt tax of NEV | The compatibility of charging INF |
| | Traffic restriction region of ICE | The lottery probability for license plate of HEV | Noise vibration and harshness of NEV | Country subsidy | The number of charging INF |
| | Traffic restriction time of HEV | The lottery probability for license plate of BEV and PHEV | Scrap value of NEV | Provincial subsidy | The distribution of charging INF |
| | Traffic restriction region of HEV | Purchase qualification of NEV | Battery range | Charging cost subsidy | The construction of fast charging INF |
| | Punishment standard for violating traffic restriction | List of NEV | Innovation of NEV | | The installation qualification of personal charging INF |
| | Traffic restriction policy for foreign license plate | Purchase qualification of ICE | | | |

It is difficult for the sensitivity analysis of the ABM with over 20 micro variables as bottom and market penetration as up. Thus, this paper will establish a bottom-up model between linking variables and macro variables. The idea of linking variables comes from analytical target cascading, which is a systematic effort to enable top-level design targets to be cascaded down to the lowest level of the modeling hierarchy [25]. Through replacing over 20 micro variables with 5 linking variables as the bottom, the sensitivity of the ABM will be easier to verify. Next, how to construct the relationship between the linking variables and micro variables is another critical problem. Compared with the fitting function, the best–worst scaling is more convenient, which will be presented in the following case study [26].

The range of five linking variables is specified as (0, 100). A higher value indicates that the consumers are more inclined to purchase NEVs under the influence of linking variables, and a lower value indicates the opposite. The NEVP is the index of purchase attitudes for NEVs, which is also limited to (0, 100) and can be formulated as a linear function of five linking variables as follows:

$$NEVP = c_1 \cdot TR + c_2 \cdot PR + c_3 \cdot PC + c_4 \cdot GS + c_5 \cdot INF, \qquad (1)$$

where $c_i, i = 1, \ldots, 5$ stand for the weights of different linking variables.

*3.3. Interaction Rules*

Figure 1 shows the decision environment in Netlogo, in which three types of consumers interact information with each other during the process of purchasing NEVs. Yellow, red, and black represent Rookie, Veteran, and New generation on the different geographic sections, respectively. Different shades of green represent different regions, which are called patches in Netlogo.

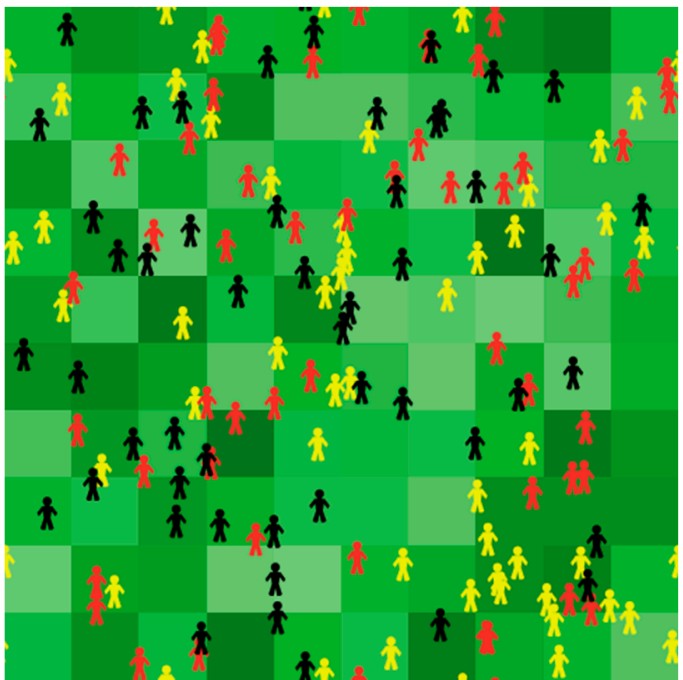

**Figure 1.** Simulation environment in Netlogo.

As shown in Figure 2, the decision process mainly involves the following six problems: (A) how long is the purchase cycle of a consumer; (B) how to initialize the value of linking variables; (C) how to define weights $c_i, i = 1, \ldots, 5$; (D) how to define the interaction range among different types of consumers; (E) what is the relationship between NEVPs and market penetration; and (F) how does the decision, i.e., to purchase or not to purchase, affect potential consumers.

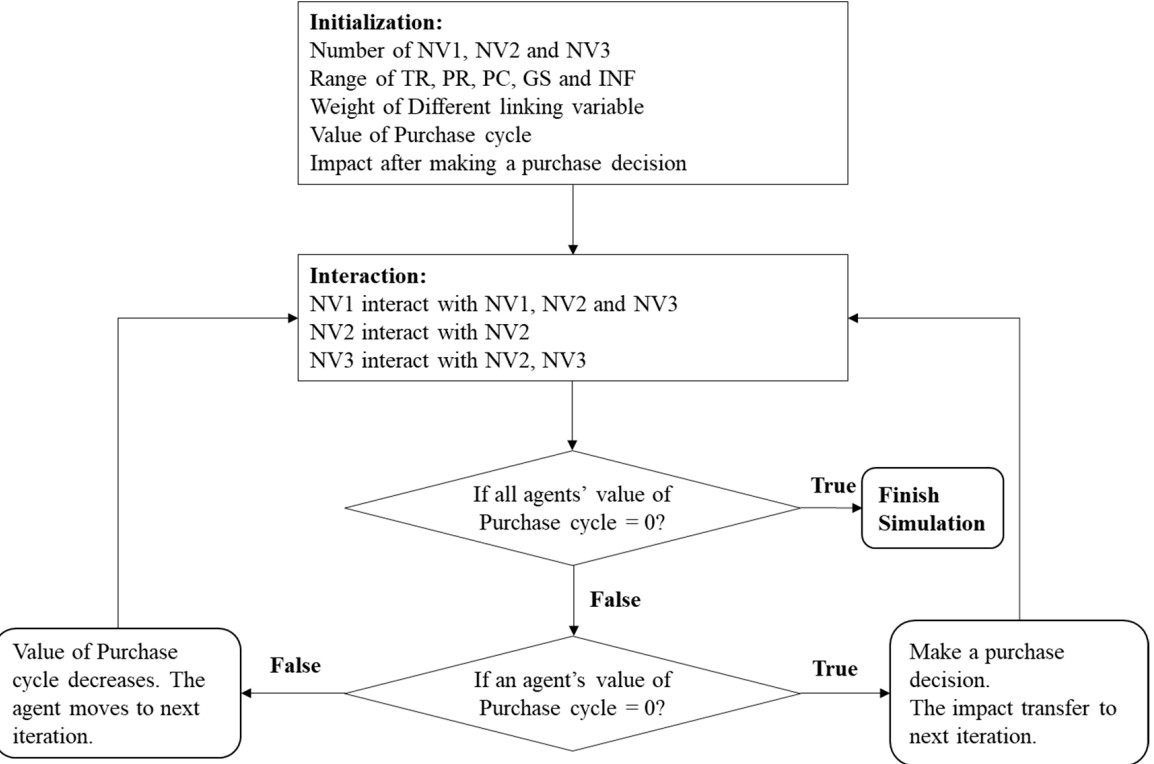

**Figure 2.** Simulation decision process of NEV purchase.

Initialization is the first step, and its specification is shown in Figure 2. The number of agents refers to the interaction ratio and the per interaction frequency. The ranges of linking variables uniformly distribute on interval values, which vary with different types of agents. The weight of linking variables is determined by a random integer. The value of the purchase cycle diminishes with iteration proceeding and becomes zero with stopping iteration. A purchase decision will leave an impact on some regions and influence adjacent agents' NEVP.

Interaction is the second step. The agents move randomly and trade NEVP with adjacent agents. Since three types of agents have different characteristics, they have different interaction ranges. The NV1 lacks a clear vehicle opinion and tends to exchange information with all nearby agents. The interact formulation of NV1 is given as the following:

$$NEVP_{self1} = NEVP_{self1} + \sum_{i=1}^{n_1} NEVP_i/2 + \sum_{i=1}^{n_2} NEVP_i/2 + \sum_{i=1}^{n_3} NEVP_i/2 ,$$
$$-(n_1 + n_2 + n_3) \cdot NEVP_{self1}/2 + E_{amount1} \tag{2}$$

where $NEVP_{self1}$ denotes the current NEVP of the NV1; $n_1, n_2, n_3$ are the number of other NV1, NV2, and NV3 on the NV1's region, respectively; $NEVP_i$ is the corresponding NEPV of some agent; and $E_{amount1}$ stands for the sum of impact of those purchase decisions on the NV1's region.

The NV2 has adequate vehicle experiences and tends to exchange information with the same type of agents. The interact formulation of NV2 is given as the following:

$$NEVP_{self2} = NEVP_{self2} + \sum_{i=1}^{n_2} NEVP_i/2 - (n_2) \cdot NEVP_{self2}/2 + E_{amount2}, \tag{3}$$

where $NEVP_{self2}$ denotes the current NEVP of the NV2; $n_2$ is the number of other NV2 on the NV2's region; $NEVP_i$ is the corresponding NEPV of some agents; and $E_{amount2}$ stands for the sum of impact of those purchase decisions on the NV2's region.

NV3 likes new things and tends to exchange information with experienced agents and the same type of agents. The interaction formulation of NV3 is given as the following:

$$NEVP_{self3} = NEVP_{self3} + \sum_{i=1}^{n_2} NEVP_i/2 + \sum_{i=1}^{n_3} NEVP_i/2 ,$$
$$-(n_2 + n_3) \cdot NEVP_{self3}/2 + E_{amount3} \tag{4}$$

where $NEVP_{self3}$ denotes the current NEVP of the NV3; $n_2, n_3$ are the number of other NV2 and NV3 on the NV3's region, respectively; $NEVP_i$ is the corresponding NEPV of some agents; and $E_{amount3}$ stands for the sum of impact of those purchase decisions on the NV3's region.

After an iteration, the termination condition will be verified. If the value of the purchase cycle equals to 0, the agent will make a purchase decision and leave an impact on its region. Table 2 shows the pseudo code for making a purchase decision. Lines 1 and 2 restrict the range of NEVP to (0, 100) under the iterations. The comparison between n4 and n5 determines the final purchase decision, i.e., purchasing a NEV or purchasing an ICE, where the range of coefficient $p$ is (0,1). The former will add $E_{variation}$ to $E_{amount}$, and the latter will add $-E_{variation}$ to $E_{amount}$. Lines 5 to 11 denote how the decision has an impact on some regions and will be transferred to the next iteration. If the value of the purchase cycle is greater than 0, the agent will move to a new location with diminishing the value of the purchase cycle and continue to iterate in the next loop.

**Table 2.** Pseudo codes for making a purchase decision.

| | |
|---|---|
| Line 1 | if NEVP > 100 [set NEVP 100] |
| Line 2 | if NEVP < 0 [set NEVP 0 ] |
| Line 3 | let n4 $p \times$ NEVP |
| Line 4 | let n5 random-float 100 |
| Line 5 | ifelse n4 >= n5 |
| Line 6 | [set Eamount Eamount + Evariation |
| Line 7 | set nev 1 |
| Line 8 | set ice 0 ] |
| Line 9 | [set Eamount Eamount—Evariation |
| Line 10 | set nev 0 |
| Line 11 | set ice 1 ] |
| Line 12 | set color grey |
| Eamount | the sum of impact on the current region |
| Evariation | a positive number |

## 4. Case Study

### 4.1. The Number of Agents

It needs to determine the size of the behavior space and the number of agents in a simulation, which will influence per interaction frequency $I_f$ and interaction ratio $I_r$. It is assumed that a square with 100 patches form the simulation space, which is 10 patches long and 10 patches wide. If N persons stay on the same patch, the maximum interaction number is $A_N^2 = N \cdot (N-1)$. Considering the different number of agents on different patches, $B_i, i = 1, 2, \ldots$ is employed to indicate the number of patches with i persons. The equations $I_f, I_r$ have the following forms:

$$I_f = \frac{\sum\limits_{i=0}^{N} B_i \cdot A_i^2}{A}, I_r = 1 - \frac{B_1}{N_p}, \tag{5}$$

where $A$ denotes the total number of agents, $A_0^2 = 0$, $A_1^2 = 0$, and $N_p$ is the total number of patches and equals to 100 in this case. We randomly distribute the different number of agents on the $10 \times 10$ patches by 10,000 times, and the average $B_i$ are presented in Table 3 and Figure 3.

**Table 3.** Interaction frequency and interaction ratio by simulation.

| | A | $B_0$ | $B_1$ | $B_1$ | $B_1$ | $B_1$ | $B_1$ | $B_1$ | $B_1$ | $B_1$ | $B_1$ | $B_1$ | $I_f$ | $I_r$ |
|---|---|---|---|---|---|---|---|---|---|---|---|---|---|---|
| 1 | 42 | 65.5 | 27.9 | 5.7 | 0.8 | 0.1 | 0.0 | 0.0 | 0.0 | 0.0 | 0.0 | 0.0 | 0.41 | 33.6% |
| 2 | 70 | 49.5 | 35.0 | 12.2 | 2.8 | 0.5 | 0.1 | 0.0 | 0.0 | 0.0 | 0.0 | 0.0 | 0.69 | 50.0% |
| 3 | 100 | 36.6 | 37.0 | 18.5 | 6.1 | 1.5 | 0.3 | 0.0 | 0.0 | 0.0 | 0.0 | 0.0 | 0.99 | 63.0% |
| 4 | 102 | 35.9 | 37.0 | 18.9 | 6.3 | 1.6 | 0.3 | 0.0 | 0.0 | 0.0 | 0.0 | 0.0 | 1.01 | 63.8% |
| 5 | 150 | 22.2 | 33.5 | 25.2 | 12.6 | 4.7 | 1.4 | 0.3 | 0.1 | 0.0 | 0.0 | 0.0 | 1.49 | 77.6% |
| 6 | 200 | 13.4 | 27.1 | 27.2 | 18.1 | 9.1 | 3.6 | 1.2 | 0.3 | 0.1 | 0.0 | 0.0 | 1.99 | 86.5% |
| 7 | 210 | 12.1 | 25.7 | 27.2 | 19.0 | 9.9 | 4.1 | 1.4 | 0.4 | 0.1 | 0.0 | 0.0 | 2.09 | 87.8% |
| 8 | 250 | 8.1 | 20.5 | 25.7 | 21.5 | 13.4 | 6.7 | 2.7 | 1.0 | 0.3 | 0.1 | 0.0 | 2.49 | 91.8% |
| 9 | 300 | 4.9 | 14.9 | 22.5 | 22.5 | 16.9 | 10.1 | 5.0 | 2.1 | 0.8 | 0.3 | 0.1 | 2.98 | 95.0% |
| 10 | 350 | 3.1 | 10.5 | 18.5 | 21.7 | 19.0 | 13.3 | 7.7 | 3.8 | 1.6 | 0.6 | 0.2 | 3.46 | 97.0% |

Figure 3 illustrates the fitted curve corresponding to the number of agents. With the increase of per interaction frequency $I_f$, the slope of the fitted curve approaches 0, and thus, the interaction ratio $I_r$ approaches 1. Leveraging the simulation efficiency and practical interaction times, we set $N_p = 150$, $N_{NV1} = 50$, $N_{NV2} = 50$ and $N_{NV3} = 50$, where $N_{NV1}$, $N_{NV2}$ and $N_{NV3}$ are the corresponding numbers of NV1, NV2, and NV3, respectively.

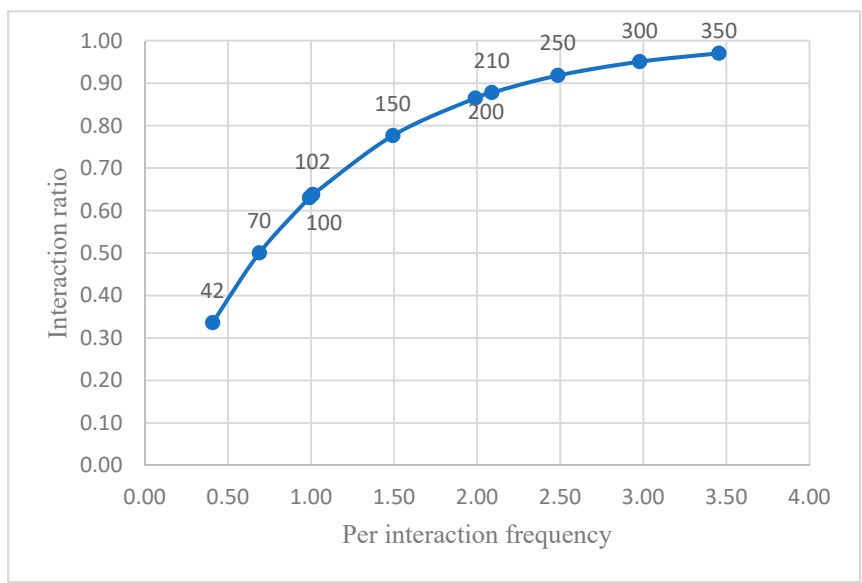

**Figure 3.** Fitted curve w.r.t. the number of agents.

### 4.2. Calibration and Verification

The coefficient $p$ is a key parameter for the purchase decision, and it is calibrated to tune up the model for fitting the real-world data. Table 4 shows the real ICE to NEV sales ratios from 2017 to July 2020. It can be seen that 5% is a relatively stable market penetration for NEVs.

**Table 4.** Vehicle sales from 2017 to July 2020.

|  | **2017** | **2018** | **2019** | **By July 2020** |
|---|---|---|---|---|
| PHEV | 102,785 | 255,545 | 179,797 | 97,037 |
| BEV | 453,586 | 751,276 | 841,687 | 302,468 |
| HEV | 141,979 | 223,294 | 294,707 | 202,507 |
| ICE (only gasoline) | 22,750,748 | 20,882,013 | 18,609,620 | 8,293,515 |
| ICE (gasoline and HEV)/ NEV (PHEV and BEV) | 41.1 | 21.0 | 18.5 | 21.3 |
| NEV/(NEV + ICE) | 2.4% | 4.6% | 5.1% | 4.5% |

Table 5 shows the distribution of linking variables. NV1 prefers NEVs in TR and PR due to a lack of vehicle knowledge and experiences. As an innovation product, NV2 and NV3 are more attractive in some novel techniques of NEVs than traditional ICEs. Aiming at the same subsidy, diversified preferences will extend the range of GS. With the rapid and unbalanced development of recharging infrastructures, NV2 has a clearer and deeper understanding for the current situation, and thus the corresponding range of INF is relatively lower.

**Table 5.** The range of linking variables.

|  | **Traffic Restriction Index (TR)** | **Purchase Restriction Index (PR)** | **NEV Product Competitiveness Index (PC)** | **Government Subsidy Index (GS)** | **Infrastructure Index (INF)** |
|---|---|---|---|---|---|
| NV1 | (60, 100) | (60, 100) | (0, 40) | (10, 90) | (40, 80) |
| NV2 | (10, 30) | (0, 20) | (40, 80) | (0, 60) | (0, 40) |
| NV3 | (30, 70) | (40, 80) | (20, 60) | (0, 80) | (20, 60) |

It is assumed that these intervals of linking variables obey a uniform distribution. Then, the mathematical expectation of NEVP with the average weight $c_i = 0.2$ can be obtained as listed in Table 6.

**Table 6.** The expected NEVP with no interaction.

|  | Expected TR | Expected PR | Expected PC | Expected GS | Expected INF | Expected NEVP |
|---|---|---|---|---|---|---|
| NV1 | 80 | 80 | 20 | 50 | 60 | 58 |
| NV2 | 20 | 10 | 60 | 30 | 20 | 28 |
| NV3 | 50 | 60 | 40 | 40 | 40 | 46 |

We compare the results under different rules, i.e., the no-interaction rules and the given interaction rules, to determine the value of the mathematical expectation method and ABM, as shown in Table 7. Compared with a real NEV market penetration of 5%, the two results are 8.8% and 4.2%, respectively, which are reasonable simulation values with $p = 0.2$. Based on the above analysis, all parameters have been determined, and we will then discuss different results of the ABM under various conditions in the following subsection.

**Table 7.** The comparison results with $10 \times 10$ patches.

| | No-Interaction Rules | | Given Interaction Rules | |
|---|---|---|---|---|
| $p = 0.2$ | Expected Number | Expected Market Penetration of NEV | Expected Number | Expected Market Penetration of NEV |
| NV1 = 50 | 5.8 | 11.6% | 2.51 | 5.0% |
| NV2 = 50 | 2.8 | 5.6% | 1.55 | 3.1% |
| NV3 = 50 | 4.6 | 9.2% | 2.26 | 4.5% |
| SUM = 150 | 13.2 | 8.8% | 6.3 | 4.2% |

*4.3. Results and Discussions*

The baseline scenario, denoted as V0, is shown in Table 8. The overall market penetration is 4.2% in Table 7, which is obtained by averaging after 10,000 calculations. The impact of the following changes will be simulated:

V1. The purchase decision does not participate in the interaction loop. Evariation = 0 indicates that there is no impact left on the patch;

V2. Increase the upper bound of NEVP to 500, which indicates that an agent has a 100% probability to purchase a NEV with $p = 0.2$;

V3. Increase the value of Evariation, which means that the impact of the purchase decision will increase;

V4. Change interaction rules;

V4-1. Each type of agent can only interact with the same type of agent;

V4-2. Each type of agent can interact with any type of agent;

V4-3. Each type of agent can only interact with NV2;

V5. Increase the number of agents, which can increase the interaction frequency.

To evaluate their possible impacts, different combinations are simulated 10,000 times, and their outcomes are averaged across all runs.

**Table 8.** The characteristics of baseline scenario V0.

|  | Description |
|---|---|
| Range of NEVP | (0, 100) |
| Interaction rules | NV1 interacts with NV1, NV2, and NV3 |
|  | NV2 interacts with NV2 |
|  | NV3 interacts with NV2 and NV3 |
| Initialization | Purchase cycle minus 1 if the agent enters into the next loop |
|  | Eamount = 0; Evariation = 10; $p = 0.2$ |
|  | Uniform distribution in TR, PR, PC, GS and INF |
|  | Purchase cycle is a random integer in [1, 6] |
|  | Weight $c_i = c_i/(c_1 + c_2 + c_3 + c_4 + c_5)$, $c_i$ is a random integer in [1, 3], $i = 1,2,3,4,5$ |
|  | NV1 = 50, NV2 = 50, and NV3 = 50 in 10×10 patches |

### 4.3.1. Sensitivity Analysis of V1 and V2

Three scenarios, i.e., V0 + V1, V0 + V2, and V0 + V1 + V2, are compared with V0, as shown in Figure 4. V0 + V1 means that Evariation = 0 substitutes Evariation = 10. V0 + V2 means that the range (0, 500) substitutes the initial range (0, 100) of NEVP. V0 + V1 + V2 means that Evariation = 0 and the range (0, 500) substitutes Evariation = 10 and the range (0, 100), respectively.

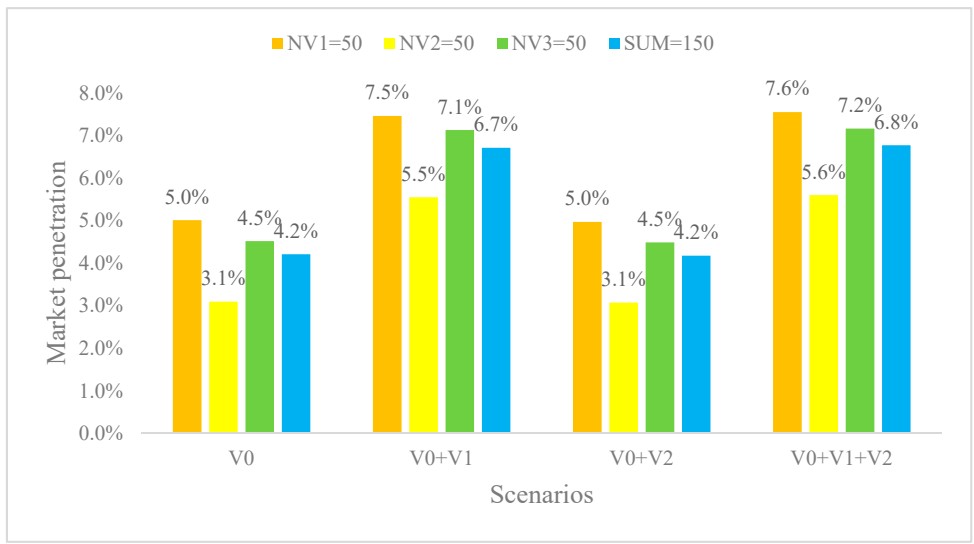

**Figure 4.** The market penetration under four scenarios V0, V0 + V1, V0 + V2, and V0 + V1 + V2.

Considering the variate V1 in two comparisons, i.e., V0 and V0 + V1 and V0 + V2 and V0 + V1 + V2, Figure 4 shows that V0 + V1 will increase the market penetration by 2.5 percentage points. Evariation = 0 means that consumers are not sensitive for the purchase decisions of adjacent consumers.

Considering the variate V2 in two comparisons, i.e., V0 and V0 + V2 and V0 + V1 and V0 + V1 + V2, the negligible difference indicates that NEVP is difficult to reach a large value in V0 or V0 + V1, due to the low-level market share of NEVs.

It is impractical to eliminate the impact of purchase decisions since it is not feasible and is contrary to the information age. The accumulation of NEVP will encounter the bottleneck under some interaction rules.

### 4.3.2. Sensitivity Analysis of V3

The larger the Evariation value, the greater the impact of the purchase decision of NEVs. We set five different values of Evariation to compare with the baseline scenario V0, as shown in Figure 5. The market penetration increases as the value of Evariation increases.

The 10 increments of Evariation will bring a 0.2 percent point increment in the total market penetration. Considering NEVP < 100, the change is very small with the increments of Evariation.

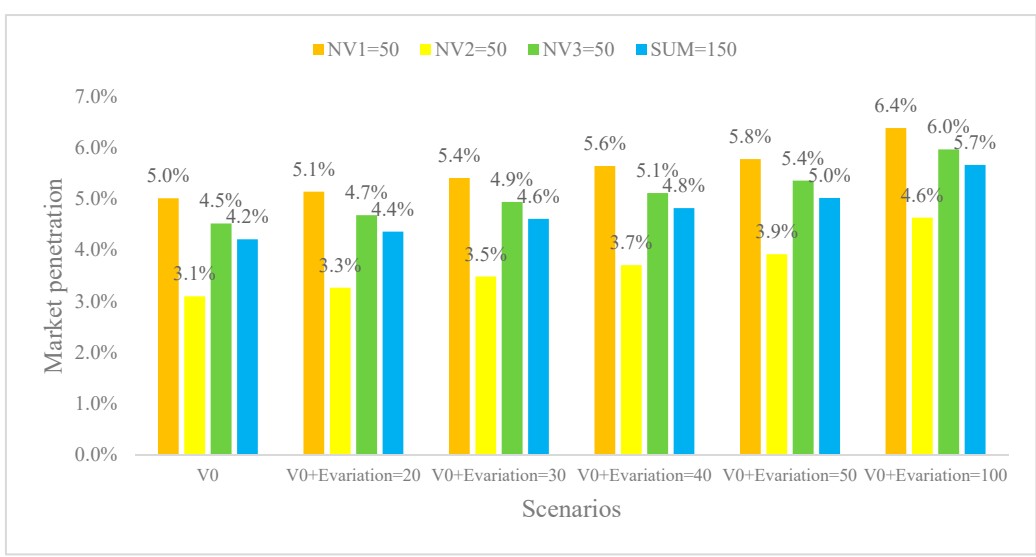

**Figure 5.** The market penetration with different values of Evariation.

For testing the impact of decreasing the residual value from ICE, we adjust (Eamount—Evariation) in Line 9 in Table 2 to (Eamount—Evariation/5). The simulated market penetration is 6.3%, and the change is obvious with 4.2% in the baseline scenario. It indicates that adjusting the purchase decision code is more effective than simply increasing the Evariation value. Purchasing NEV will increase the NEVP of adjacent agents and purchasing ICE will reduce the NEVP of adjacent agents. When the purchase of NEVs is more attractive or the purchase anxiety of NEVs is not sensitive, the increment in NEVP by purchasing NEVs will exceed the decrement in NEVP by purchasing ICEs. This change will increase the market penetration of NEVs.

### 4.3.3. Sensitivity Analysis of V4

In this section, V0 is compared with three other interaction rules, as shown in Table 9. V0 + V4.1 assumes that each type of agent prefers to interact with the same type of agent. V0 + V4.2 assumes that each type of agent can interact with any agent. V0 + V4.3 assumes that each type of agent can only interact with NV2.

**Table 9.** Three different interaction rules with V0.

| Scenario | V0 | V0 + V4.1 | V0 + V4.2 | V0 + V4.3 |
|---|---|---|---|---|
| Interaction rules | NV1 interacts with NV1, NV2, and NV3 | NV1 interacts with NV1 | NV1 interacts with NV1, NV2, and NV3 | NV1 interacts with NV2 |
| | NV2 interacts with NV2 | NV2 interacts with NV2 | NV2 interacts with NV1, NV2, and NV3 | NV2 interacts with NV2 |
| | NV3 interacts with NV2, NV3 | NV3 interacts with NV3 | NV3 interacts with NV1, NV2, and NV3 | NV3 interacts with NV2 |

The simulated results are shown in Figure 6. These three assumptions are formed based on some practical situations according to preferential attachment theory [27]. V0 + V4.1 means that some people just tend to believe in men with similar attitudes. V0 + V4.2 means that some people are open-minded and like to socialize with all the adjacent people. V0 + V4.3 means that someone just tends to believe in authority, such as experienced people. The overall market penetration of V0 + V4.1 and V0 + V4.2 is 6.0%, but

the former has a bigger variance among the three types of agents. The simulated results of V0 and V0 + V4.3 are similar, except for the tiny increase in NV1′s market penetration in V0 + V4.3. V0 + V4.2 is the most balanced rule and brings increment to three types of agents.

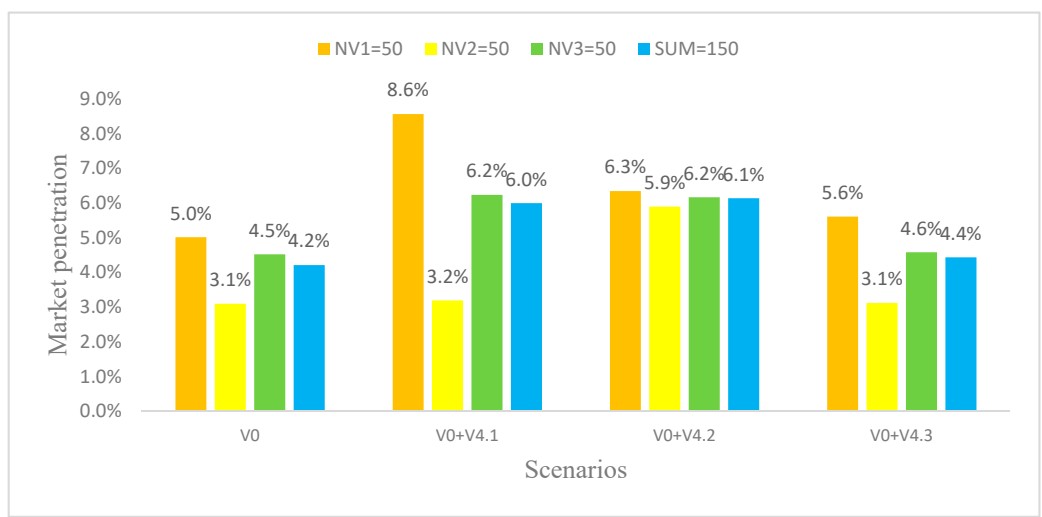

**Figure 6.** The market penetration with different interaction rules.

How to expand the interaction scope? First, some high traffic automobile websites can provide purchase exchange groups for potential consumers. In the internet age, most people would browse some automobile information: prices, configurations, and using experiences, etc. Compared with the traditional social relationship, online forum, or group is a good communication platform. Second, automobile stores can establish a Wechat group for their potential consumers and answer some questions in time. Traditional sales pattern generally avoids unnecessary contact among consumers and worries that vague price or information will bring troubles. With some automobile Apps, many automobile information is open, especially for NEVs dealers. Different types of potential consumers can exchange information online through Wechat, a necessary communication App in China.

### 4.3.4. Sensitivity Analysis of V5

We set a scenario V5: $N_p = 300, N_{NV1} = 100, N_{NV2} = 100$ and $N_{NV3} = 100$. The population number and the per interaction frequency are twice the corresponding values in V0. Figure 7 shows the comparison results between V0 + V5 and V0 + V5 + V4.2.

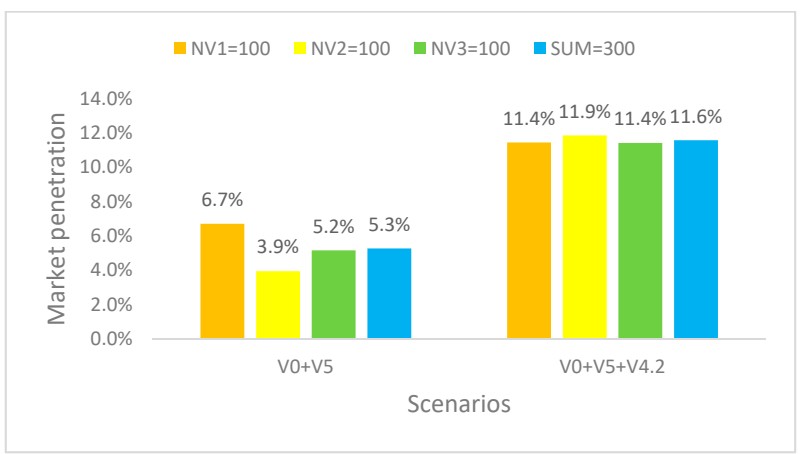

**Figure 7.** The market penetration with different population density.

Increasing the per interaction frequency can increase the market penetration. It is a more effective and direct way by promoting NEVs in cities with high car ownership. In addition to the high-density population, the interaction frequency is also related to the personality of consumers. For example, experienced automobile consumers generally have relatively fixed social relationships, and thus it is more practical by increasing their interaction frequency. The inexperienced potential consumers generally have some hesitations during the purchase process, and thus it is easier for increasing their interaction scope. Interaction frequency and interaction scope have some similarities, but they belong to different dimensions. The V0 + V5 + V4.2 has the highest market penetration at 11.6% and shows a significant joint effect.

### 4.4. Establish the Relationship between Linking Variables and Micro Variables

Regression function, survey questionnaire, or discussion can be used to investigate the relationship between linking variables and micro variables. Owing to the rapid development of NEVs, it is difficult to describe the relationship by using an accurate functional relationship, and thus the best–worst scaling (BW) is employed to describe the relationship [28]. BW avoids the problem of rating bias, and it is powerful in conducting cross cultural studies in consumer behavior [29]. An example of linking variable PR is presented. Six influencing factors of PR are listed in Table 1, and the corresponding balanced incomplete blocks designs are shown in Table 10.

**Table 10.** Balanced incomplete blocks designs for six factors.

| Choice Set Number | (6,5,2,2) | | | The Number of the Attribute | The Corresponding Description |
|:---:|---|---|---|:---:|:---:|
| (1) | 1 | 2 | 5 | 1 | The lottery probability for license plate of ICE |
| (2) | 1 | 2 | 6 | 2 | The lottery probability for license plate of HEV |
| (3) | 1 | 3 | 4 | 3 | The lottery probability for license plate of EV and PHEV |
| (4) | 1 | 3 | 6 | 4 | Purchase qualification of NEV |
| (5) | 1 | 4 | 5 | 5 | List of NEV |
| (6) | 2 | 3 | 4 | 6 | Purchase qualification of ICE |
| (7) | 2 | 3 | 5 | | |
| (8) | 2 | 4 | 6 | | |
| (9) | 3 | 5 | 6 | | |
| (10) | 4 | 5 | 6 | | |

Moreover, (10, 5, 3, 2) means that there are 10 choice sets for 6 factors, each factor appears 5 times across all choice sets, each choice set contains 3 factors, and each factor appears twice with each other. Table 11 presents a questionnaire of one choice set, i.e., the choice set number (1) in Table 10. The BW score and the average BW score can be derived through data analysis to compare the attribute importance.

**Table 11.** An example of a BW choice set as presented to respondents.

| Remember the Purchase Restriction Index. For Each of the Following Tables, Tick the ONE Reason that MOST Influenced Your Choice and the ONE that LEAST | | | |
|:---:|:---:|:---:|:---:|
| **Least/Worst** | | **Attribute** | **Most/Best** |
| ☐ | 1 | The lottery probability for license plate of ICE | ☐ |
| ☐ | 2 | The lottery probability for license plate of HEV | ☐ |
| ☐ | 3 | List of NEV | ☐ |

Authors should discuss the results and how they can be interpreted from the perspective of previous studies and of the working hypotheses. The findings and their implications should be discussed in the broadest context possible. Future research directions may also be highlighted.

## 5. Implications of Our Work

### 5.1. Theoretical Implication

Little research pays attention to the tradeoff between granularity and accuracy. Compared with decomposing a large-scale ABM or simplifying the interaction rules, our research develops an agent-based model with linking variables (ABML) in solving the dilemma between microlevel and macrolevel.

Our work gives a three-level variables structure including micro, linking, and macro variables. The emergence from bottom to top occurs between linking and macro variables, and the best–worst scaling describes the mapping between micro and linking variables. By introducing the population density and the interaction frequency, we determine the number of agents with an experiment.

The ABML is a systematic method, which can be studied further or duplicated in the market penetration of the new energy vehicles. Though our work is aimed at China, these management insights are still valid in other countries:

(1) This paper develops an agent-based model with linking variables (ABML) to investigate the factors affecting the new energy vehicles market in China. The ABML is a general framework and provides causal insights among knowledge (micro variables), association (linking variables), and conception (macro variables).

(2) The emergence from bottom to top occurs between the linking and macro variables, and the best–worst scaling describes the mapping between the micro and linking variables. This method can alleviate the overburden variables and the complexity of the simulation.

(3) The agent choice description is newly developed. Rookie, Veteran, and New Generation consumers are suitable for China's market. The categorization of agents is helpful for the interaction and can be applied to some similar research.

(4) This paper creatively proposes an experimental method to determine the size of the behavior space and the number of agents, which has two evaluation dimensions, i.e., the per interaction frequency and the interaction ratio.

### 5.2. Managerial Implication

This paper presents a detailed and complete case study and obtains some management insights as follows:

(1) The low-level market share of NEVs will inhibit the accumulation of purchase intention and product competitiveness in NEVs. Market penetration is an important index for evaluating the development of NEVs.

(2) Increasing the purchase attraction of NEVs or weakening the purchase anxiety of NEVs can narrow the difference between the increment in NEVP by purchasing NEVs and the decrement in NEVP by purchasing ICEs. The change will bring the market penetration of NEV a foreseeable increment.

(3) Expanding the scope of consumers' interaction can increase the total market penetration of NEVs, including browsing automobile websites and establishing Wechat groups. The open and innovative marketing model has been validated in Tesla.

(4) It is more effective by investing and promoting NEVs in cities with high car ownership. The rapid development of NEVs in Beijing and Shanghai also confirms this point. These cities have more advantages for NEVs in traffic restriction, purchase restriction, government subsidy, and infrastructures.

(5) In the technological dimension, NEVs are technology-intensive innovative products, while ICEs have been mature products for over 100 years. Considering the unbalanced distribution of charging infrastructures and low charging efficiency, it must rely on developing unique characters and usage scenarios to increase the market penetration of NEVs, and thus innovation is important for NEVs.

## 6. Conclusions

This paper aims to develop an agent-based model with linking variables (ABML) to analyze China's NEVs market. According to the characteristics of Chinese consumers, consumers are divided into three types of agents: Rookie, Veteran, and New Generation. The model variables are categorized into three levels: micro variables, linking variables and macro variables. This paper uses five linking variables as the bottom and the market penetration as the top in ABML, where NEVP reflects the results of the interactions. The relationship between micro variables and linking variables is connected by the best–worst scaling. The ABML considers different purchase cycles and introduces a residual rule to update NEVP. Some other scenarios are compared with the baseline scenario in sensitivity analysis, and some following conclusions are obtained. The proposed ABML establishes a general framework for solving similar simulation problems with too many variables and complex functional relationships. Through adjusting the corresponding variables and establishing the corresponding rules, this framework can be applied to many other social experiment scenarios.

This study on the market penetration of NEVs in China has raised several further research directions. First, the fitting function between linking variables and macro variables is assumed to be a linear function of part-worth values in this paper. An alternative approach is adopting nonlinear functions such as the Multinomial Logit choice model [30]. Second, the weights of the linking variables are assumed to be random integers. An efficient weighting update method can be applied to the iteration process [31]. Third, the sensitivity analysis can be further studied. In this case, the number of agents is averaged. However, the number of agents and the location of agents are uneven in some regions. In addition to the population density and the interaction frequency, geographic information can also be introduced for the analysis of the population distribution. Fourth, the stakeholder engagement can encourage the spread of NEVs, and a multi-stakeholder approach can enrich the agent choice of ABML in the future [32]. Fifth, the proposed method can be applied in some related domains such as commercial vehicles, the food supply chain, and new ecological paradigms [33].

**Author Contributions:** Writing—original draft, M.C.; writing—review and editing, R.X.J.L.; visualization, C.L. All authors have read and agreed to the published version of the manuscript.

**Funding:** This research was funded by Humanities and Social Sciences Youth Foundation of Ministry of Education of China, Grant/Award Number: 19YJCZH009.

**Institutional Review Board Statement:** Not applicable.

**Informed Consent Statement:** Not applicable.

**Data Availability Statement:** No new data were created or analyzed in this study.

**Acknowledgments:** We sincerely thank Xiaoliang Zhang for some insightful articles on the https://www.so.car/ (accessed on 6 July 2021)), who is the CEO of the SoCar car data workshop. We sincerely thank the editor and three reviewers for their kind and helpful comments on this paper.

**Conflicts of Interest:** We declare that we have no conflicts of interest to this work.

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
