# Peer review of "How to Improve the Market Penetration of New Energy Vehicles in China: An Agent-Based Model with a Three-Level Variables Structure"

_sustainability, doi:10.3390/su132112307_

Round 1

Reviewer 1 Report

The idea of studying new energy vehicles and the ways in which its market penetration could be improve is noteworthy and worthwhile. The use of the agent-based model with a three-level variables structure is also unique and demonstrates good value in realizing the intended idea. The experimental approach is also commendable, as this enables the study to provide causal insights (third level knowledge), which is the most powerful form of knowledge after conception- (first level) and association- (second level) related knowledge. Nonetheless, the paper could be improved in several ways, which I will explain as follows.

First, I think there is a need to talk about sustainability marketing and sustainable consumption, as this study’s promotion of new energy vehicles relates to sustainability marketing (Lim, 2016), while the usage of new energy vehicle itself is a form of sustainable consumption (Lim, 2017). This can enhance the scope of contribution of the insights that are derived from the study in this paper.

Lim, W. M. (2016). A blueprint for sustainability marketing: Defining its conceptual boundaries for progress. Marketing Theory, 16(2), 232-249.

Lim, W. M. (2017). Inside the sustainable consumption theoretical toolbox: Critical concepts for sustainability, consumption, and marketing. Journal of Business Research, 78, 69-80.

Second, the agent choice description in Section 3.1 is interesting. Is this new or is this adopted from somewhere? If it is adopted, then this needs to be cited. If it is newly developed, then some form of validation (e.g., pre-test) is needed, and it can then be mentioned as a contribution of the present study too.

Third, the implications of the research needs to be separated into sub-sections: theoretical implications and managerial implications.

Fourth, a conclusion section should be added, with the first paragraph outlining the key takeaways and the second paragraph dedicated to limitations and future research directions.

Good luck and all the best!

Author Response

Comments and Suggestions for Authors

The idea of studying new energy vehicles and the ways in which its market penetration could be improve is noteworthy and worthwhile. The use of the agent-based model with a three-level variables structure is also unique and demonstrates good value in realizing the intended idea. The experimental approach is also commendable, as this enables the study to provide causal insights (third level knowledge), which is the most powerful form of knowledge after conception- (first level) and association- (second level) related knowledge. Nonetheless, the paper could be improved in several ways, which I will explain as follows.

First, I think there is a need to talk about sustainability marketing and sustainable consumption, as this study’s promotion of new energy vehicles relates to sustainability marketing (Lim, 2016), while the usage of new energy vehicle itself is a form of sustainable consumption (Lim, 2017). This can enhance the scope of contribution of the insights that are derived from the study in this paper.

Lim, W. M. (2016). A blueprint for sustainability marketing: Defining its conceptual boundaries for progress. Marketing Theory, 16(2), 232-249.

Lim, W. M. (2017). Inside the sustainable consumption theoretical toolbox: Critical concepts for sustainability, consumption, and marketing. Journal of Business Research, 78, 69-80.

Response: Thanks for high-quality, constructive reviews. The two references are quite insightful for us to enhance the scope of contribution. After reading the two references carefully, we add Lim (2016) to Section 1 introduction, and Lim (2017) to Section 2.1 scope of the study. In addition, the knowledge, conception and association are very precise definitions for micro, linking and macro level. We have added them in the theoretical implication. We also add some management insights in Section 5 from the perspective of sustainability marketing and sustainable consumption.

Second, the agent choice description in Section 3.1 is interesting. Is this new or is this adopted from somewhere? If it is adopted, then this needs to be cited. If it is newly developed, then some form of validation (e.g., pre-test) is needed, and it can then be mentioned as a contribution of the present study too.

Response: Thanks for the good advice. The agent choice description is mentioned in a Chinese book, and newly developed in a research article. We have added the explanations of the agent choice in Section 3.1 and cited the book as a reference.

Third, the implications of the research needs to be separated into sub-sections: theoretical implications and managerial implications.

Response: Thanks for the advice. We have separated the implication into the theoretical section and the managerial section.

Fourth, a conclusion section should be added, with the first paragraph outlining the key takeaways and the second paragraph dedicated to limitations and future research directions.

Response: Thanks for the advice. We have added a conclusion section with two paragraphs. 

Reviewer 2 Report

The topic of the article is up-to-date. It corresponds with contemporary research trends.

The structure of the article is correct - it consists of the short theoretical descriptive part and research part, which has been drafted very precisely and concisely. The paper is logically and transparently designed.

The authors focused on the possibilities of using Agent-Based Model modelling (ABM) in the field of the market analysing. This type of modelling is particularly useful in complex systems, with dynamic links between its components, and it was very interesting to get acquainted with  the results they’ve achieved.

Author Response

Comments and Suggestions for Authors

The topic of the article is up-to-date. It corresponds with contemporary research trends.

The structure of the article is correct - it consists of the short theoretical descriptive part and research part, which has been drafted very precisely and concisely. The paper is logically and transparently designed.

The authors focused on the possibilities of using Agent-Based Model modelling (ABM) in the field of the market analysing. This type of modelling is particularly useful in complex systems, with dynamic links between its components, and it was very interesting to get acquainted with  the results they’ve achieved.

Response: Thanks for high-quality, constructive reviews. We have polished the paper throughout.

Reviewer 3 Report

In the assessment of the paper submitted for the review, I specifically focussed on the discussed issues, applied research methods and the scope of analysis of research results, as well as substantive content of the article and its structure.

The considerations conducted in the paper are focused on such categories as: new energy vehicles, agent-based model, linking variables, market penetration.

The reviewed paper is of scientific nature.

The subject area discussed in the paper is important.

The article, however, needs to be refined to be published.

Therefore, it is specifically recommended to:

- develop the literature review,

- take into consideration other latest publications in the sphere of discussed subject matter,

-deepen the discussion,

- indicate the research limitations and trends for future research.

Author Response

Comments and Suggestions for Authors

In the assessment of the paper submitted for the review, I specifically focussed on the discussed issues, applied research methods and the scope of analysis of research results, as well as substantive content of the article and its structure.

The considerations conducted in the paper are focused on such categories as: new energy vehicles, agent-based model, linking variables, market penetration.

The reviewed paper is of scientific nature.

The subject area discussed in the paper is important.

The article, however, needs to be refined to be published.

Therefore, it is specifically recommended to:

- develop the literature review

Response: Thanks for the good advice. We have separated the literature review into two sub-sections, including scope of the study and research areas of ABM.

- take into consideration other latest publications in the sphere of discussed subject matter

Response: Thanks for the good advice. We have added 6 new references and substituted 4 latest references for old references.

-deepen the discussion

Response: Thanks for good pint. We have reconstructed the discussion, where theoretical implication and managerial implication are added.

- indicate the research limitations and trends for future research

Response: Thanks for the advice. We have added a conclusion section with two paragraphs, with the first paragraph outlining the key takeaways and the second paragraph dedicated to limitations and future research directions.

Round 2

Reviewer 3 Report

Thanks for having taken into account my comments.   Best wishes.

Author Response

Thanks for high-quality, constructive reviews. We have polished the paper throughout.